# Erythrocyte Plasma Membrane Lipid Composition Mirrors That of Neurons and Glial Cells in Murine Experimental In Vitro and In Vivo Inflammation

**DOI:** 10.3390/cells12040561

**Published:** 2023-02-09

**Authors:** Agnese Stanzani, Anna Sansone, Cinzia Brenna, Vito Antonio Baldassarro, Giuseppe Alastra, Luca Lorenzini, Chryssostomos Chatgilialoglu, Ilaria Laface, Carla Ferreri, Luca Maria Neri, Laura Calzà

**Affiliations:** 1Department of Pharmacy and Biotechnology (FaBiT), University of Bologna, 40126 Bologna, Italy; 2Institute for Organic Synthesis and Photoreactivity (ISOF), National Research Council (CNR), 40129 Bologna, Italy; 3Department of Translational Medicine, University of Ferrara, 44121 Ferrara, Italy; 4LTTA—Electron Microscopy Center, University of Ferrara, 44121 Ferrara, Italy; 5Department of Veterinary Medical Sciences (DIMEVET), University of Bologna, Ozzano dell’Emilia, 40064 Bologna, Italy; 6Health Science and Technologies, Interdepartmental Center for Industrial Research (HST-ICIR), University of Bologna, Ozzano dell’Emilia, 40064 Bologna, Italy

**Keywords:** lipids, fatty acids, cholesterol, GM1, lipid rafts, experimental allergic encephalomyelitis, inflammation

## Abstract

Lipid membrane turnover and myelin repair play a central role in diseases and lesions of the central nervous system (CNS). The aim of the present study was to analyze lipid composition changes due to inflammatory conditions. We measured the fatty acid (FA) composition in erythrocytes (RBCs) and spinal cord tissue (gas chromatography) derived from mice affected by experimental allergic encephalomyelitis (EAE) in acute and remission phases; cholesterol membrane content (Filipin) and GM1 membrane assembly (CT-B) in EAE mouse RBCs, and in cultured neurons, oligodendroglial cells and macrophages exposed to inflammatory challenges. During the EAE acute phase, the RBC membrane showed a reduction in polyunsaturated FAs (PUFAs) and an increase in saturated FAs (SFAs) and the omega-6/omega-3 ratios, followed by a restoration to control levels in the remission phase in parallel with an increase in monounsaturated fatty acid residues. A decrease in PUFAs was also shown in the spinal cord. CT-B staining decreased and Filipin staining increased in RBCs during acute EAE, as well as in cultured macrophages, neurons and oligodendrocyte precursor cells exposed to inflammatory challenges. This regulation in lipid content suggests an increased cell membrane rigidity during the inflammatory phase of EAE and supports the investigation of peripheral cell membrane lipids as possible biomarkers for CNS lipid membrane concentration and assembly.

## 1. Introduction

Lipids and proteins are the predominant structural elements of all cell membranes, which also define the functional subcompartments at micron, submicron and nanometer scales. Neurons, astrocytes, microglial cells, oligodendrocytes (OLs) and OPCs are the most common cells in the central nervous system (CNS), all being highly specialized cells with lipid-defined membrane macrodomains, such as the synaptic membrane in neurons, or the multilayered myelin sheath and nodal domains in OLs, the myelinating cells of the CNS [1].

The lipid composition in the CNS differs substantially between gray and white matter, mainly due to the respective content of myelin, the specialized multilamellar membrane consisting of 40 or more tightly wrapped lipid bilayers formed by the OL plasma membrane. The myelin sheath is characterized by a high lipid content (70–85%), while most biological membranes have an approximatively equivalent ratio of proteins to lipids (50–50%) [2]. Myelin membrane lipids belong to all major classes of lipids, including cholesterol (>25% by weight), glycosphingolipids (especially galactosylceramide) and ether-glycerophospholipids [3,4], with an approximate 2:2:1 ratio of cholesterol, phospholipids (mainly ethanolaminephosphatide and phosphatidylcholine) and glycolipids (galactosylceramide and sulfatide). All these lipids, except for cholesterol, share the use of FAs as building blocks [5].

White matter integrity requires continuous myelin synthesis and turnover [3,6], and the biosynthesis, storage and cellular trafficking of lipids are all essential for the assembly and maintenance of myelin through its life span [6]. In particular, myelin requires a high amount of FAs for its assembly and maintenance, and myelinating cells are particularly vulnerable to FA and lipid disorders [7]. Mature OLs undergo a slow but constant physiological turnover of their myelin sheath, thereby establishing an equilibrium of replenishment and degradation that can also be subject to neural activity-related adaptations, with consequences for nerve conduction velocity and associated neurophysiological functions [8]. Notably, this turnover is finely regulated, affecting certain characteristics, such as the myelin sheath length dynamics, but not others, such as the myelin sheath thickness [9].

Brain lipid composition and turnover have also emerged as major players in CNS lesions and diseases in light of both the possible role of altered lipid availability for lipid membrane turnover and bioactivities and the influence of pathogenic events on the lipid membrane structure and bioactive lipid activity [10,11]. Myelin turnover is critical for the repair of myelin damage, a process that depends on a variety of factors, such as damage severity, age and nutritional status, the latter requiring a fine balance between an adequate dietary lipid intake [12] and the potentially toxic effects of excess lipids [13].

Inflammation is a pathogenic event shared by many inflammatory diseases, such as vascular and traumatic injuries, neurodegenerative diseases, such as Alzheimer’s disease, and primary white matter pathologies, such as multiple sclerosis (MS). Inflammation has a dual impact on lipids. On the one hand, certain lipids, such as sphingolipid metabolites, signal molecules and regulate a diverse range of cellular processes that play a key role in immunity, inflammation and inflammatory disorders. The role of the balance between the membrane phospholipid content of the PUFA omega-3 docohexaenoic acid (DHA) and the PUFA omega-6 arachidonic acid (ARA) has been clearly linked to the release of FAs and their conversion to mediators upon receptor-mediated signal transduction in neurons and glial cells, implicated in excitotoxicity, inflammation and ischemia [14]. On the other hand, imbalances in the level and metabolism (β-oxidation, synthesis, desaturation, elongation and peroxidation) of FAs drive the initiation and progression of CNS disorders [15].

Finally, dietary regimes with a focus on lipid intake, such as the ketogenic diet and diets high in FAs, are under active investigation as a form of adjuvant therapy in neurological diseases [16], while the role of dietary intervention in MS, the most common inflammatory demyelinating disease, is still controversial [17].

In this complex scenario, the use of peripheral lipids as “biomarkers” to reflect the lipid composition of brain membranes is an attractive and challenging objective, also in view of the “Holy Grail” in the field—the search for a peripheral, noninvasive and robust biomarker reflecting CNS composition. In this study, we approached this goal at a preclinical level using biochemical techniques (for FA composition) and microscopy (for cholesterol, GM1 and membrane fluidity fluorescence probes) to compare peripheral erythrocytes (RBCs) to CNS tissues obtained from mice at different stages of EAE, the most widely used animal model for MS. EAE is characterized by intense inflammation and the extended demyelination of the spinal cord, following a defined clinical course that includes an acute and a remission phase. To investigate lipids in the plasma membrane of individual cell types in the CNS, we used primary CNS cells (neurons, OPCs and mature oligodendrocytes) acutely exposed to inflammatory challenges.

## 2. Materials and Methods

### 2.1. Animals and Study Design

Forty-four C57BL/6, 10-week-old female mice (Charles River Laboratories, Calco, Lecco, Italy) were used in this study, housed three per cage on a 12 h light/dark cycle, with food and water ad libitum. All animal protocols described herein were carried out in accordance with the European Community Council Directives (2010/63/UE), were approved by the Ministry of Health (no. 635/2018-PR) and complied with the guidelines published in the NIH and ARRIVE Guide for the Care and Use of Laboratory Animals.

Three groups of mice were included in the study: the control group (CTRL), the EAE acute phase group (mice sacrificed at 14 days postimmunization, DPI) and the EAE remission phase group (mice sacrificed at 28 DPI). The number of animals included in each group was calculated using a priori power analysis with α = 0.05, (1−β) = 0.08 and d = 1.15, considering the clinical values described in the literature and the minimum number of mice needed at the remission phase (N = 13).

On the day of immunization (Day 0), the mice received two MOG35–55 (AS-60130-5, AnaSpec, Fremont, CA, USA) emulsion subcutaneous injections (0.05 mL/injection), one near the left axillary lymph nodes and one near the left inguinal lymph nodes. A pertussis toxin (PTx Bottle M Tuberculosis H37Ra Desiccated; 231141, BD, Franklin Lakes, NJ, USA; 500 ng/injection) was also injected intraperitoneally (0.30 mL/injection). On day 2, mice received another PTx injection. On day 7, each mouse received another two MOG35-55 emulsion injections (sc, 0.05 mL/injection) in the right axillary and inguinal lymph node areas.

The mice were weighed daily and examined for clinical signs of EAE according to the following semiquantitative score for neurological disability: 0 = no signs; 1 = loss of tail tone; 2 = mono- or bilateral weakness of hind legs or middle ataxia; 3 = ataxia or paralysis; 4 = severe hind leg paralysis; 5 = severe hind leg paralysis and urinary incontinence. Wet food was included inside the cages to facilitate feeding due to the animals’ disability. Eight EAE mice were sacrificed for a tissue lipidomic analysis and neuropathological characterization in the acute phase. Thirteen were sacrifices at 55 DPI, when blood and tissues were collected for the tissue lipidomic analysis and neuropathological characterization at the remission phase. Blood sampling was also performed in all animals at 20 DPI.

Blood was collected from the control group, EAE acute phase group and EAE remission phase group in Eppendorf tubes containing K3-EDTA, and the red blood cells (RBCs) were analyzed in less than 5 h for scan electron microscopy and lipidomic. In addition, N = 8 mice were used for an EAE neuropathological characterization.

### 2.2. Histology and Immunofluorescence

On the day of sacrifice, the mice were euthanized with an overdose of anesthetic and perfused with 4% paraformaldehyde and 14% picric acid in 0.2 M Sorensen buffer (pH 6.9). The mice’ spinal cords were postfixed for 24 h, then washed in 5% sucrose in 0.2 M Sorensen buffer (pH 7.4), while the lumbar spinal cord was frozen in N_2_ and kept at −20 °C until processing. Sections of the lumbar spinal cords (14μm thickness) were cut on a Leica CM1950 cryostat (Leica Byosystems Walldorf, Germany).

Toluidine blue 0.1% staining was performed on dried slides. Sections were then hydrated in 100% alcohol for 20′’, 90% alcohol for 20′’ and 70% alcohol for 20′’, and then dyed in toluidine blue for 6′. After staining, samples were dehydrated in 70% alcohol for 20′’, 90% alcohol for 20′’ and 100% alcohol for 20′’, 5′ in xylene, and then mounted using Eukitt^®^ quick-hardening mounting medium (Sigma-Aldrich, St. Louis, MO, USA).

IF was performed on a contiguous slide that had been dried and then rehydrated with phosphate-buffered saline (PBS) 0.01 M pH 7.4. After rehydration, the samples were incubated for 1 h at room temperature with a preabsorption solution consisting of donkey serum 1% and BSA 1% in PBS 0.01 M pH 7.4. After preabsorption, the samples were incubated with primary antibodies diluted in Triton 0.3% in a PBS 0.01 M pH 7.4 solution overnight at 4 °C. The primary antibody dilutions used were rabbit antimyelin basic protein (MBP) (1:250 Proteintech; Rosemont, IL, USA), rabbit antiglial fibrillary acid protein (GFAP) (1:500, Dako), mouse anti-neurofilament 200 (NF2009 (1:200, Sigma)) and rabbit anti-IBA1 (1:250, Wako Chemicals; Richmond, VA, USA). After rinsing in PBS, the sections were incubated at 37 °C for 2 h with the secondary antibodies diluted in Triton 0.3% in a PBS 0.01 M pH 7.4 solution. The secondary antibody dilutions used were donkey antirabbit IgG Cy2 (1:200, Jackson Immunoresearch; West Grove, PA, USA) and donkey antimouse IgG RRX (1:200, Jackson Immunoresearch). The sections were then rinsed in PBS and mounted in a phenylenediamine solution. Control slices were incubated with the secondary antibodies only and processed in parallel. Images were captured using a Nikon Eclipse E600 microscope equipped with a digital CCD camera Q Imaging Retiga-2000RV (Q Imaging, Surrey, BC, Canada) equipped with the NIS software v4.30.02 (Nikon, Tokyo, Japan).

### 2.3. Lipid Analysis

Commercially available cis and trans fatty acid methyl esters (FAMEs) were purchased from Merck (Darmstadt, Germany); trans FAMEs not commercially available were synthetically obtained as described [18,19,20]; 6-cis C16:1, 7 cisc16:1, 8 cis. 18:1, cis 5 and cis 8 18:2 acid methyl esters were commercially available from Lipidox (Lidingö; Sweden); chloroform, methanol, diethylether, n-hexane (HPLC grade) and PBS were purchased from Baker (Hoboken, NJ, USA) and used without further purification. Analytical thin-layer chromatography (TLC) was performed on Merck silica gel 60 plates (0.25 mm thickness), and spots were detected by spraying the plate with a cerium ammonium sulfate/ammonium molybdate reagent and revealed by heating the plate.

FAMEs were analyzed with GC (6850; Agilent Dako, Santa Clara, CA, USA) using the split mode (50:1) equipped with a 60 m × 0.25 mm × 0.25 mm (50% cyanopropyl) methylpolysiloxane column (DB23; Agilent) and a flame ionization detector with the following oven program: temperature initially set at 165 °C, held for 3 min, followed by an increase of 1 °C/min up to 195 °C, held for 40 min, followed by a second increase of 10 °C/min up to 240 °C, held for 10 min. A constant pressure mode (29 psi) was chosen with helium as the carrier gas. FAMEs were identified by comparison with the retention times of authentic samples and expressed in quantitative relative percentages (mean ± SD), quantified on the basis of standard reference calibration curves [21,22]. A representative gas chromatogram is shown in Appendix A.

The work-up of the blood samples followed the described procedures [23,24,25]. In brief, each whole blood sample (100 µL) in EDTA was added to PBS (0.5 mL) and centrifuged at 5000× *g* for 10 min at 4 °C for 5 consecutive cycles to eliminate any traces of plasma. RBC lysis was performed by adding tridistilled H20 (1 mL), followed by centrifugation at 15,000 for 15 min to isolate the RBC membrane pellets, which were then reconstituted in H2O (1 mL) and added to 2:1 chloroform/methanol (2 × 4 mL) for lipid extraction [26]. The organic layers were collected and dried over anhydrous Na2SO4 and evaporated under vacuum to dryness. The lipid extracts (0.64 ± 0.07 mg), consisting of phospholipids and cholesterol, as revealed using TLC (n-hexane-diethyl ether 9:1), were converted to FAMEs through the addition of 0.5 mL of KOH/MeOH 0.5 M and stirring the solution at room temperature for 10 min. The reaction was then quenched with brine (0.5 mL) and the FAMEs extracted with n-hexane (4 × 2 mL); the organic layers were collected, dried over anhydrous Na_2_SO_4_ and evaporated to dryness. The FAME residue of each sample was dissolved in 10 µL of n-hexane and 1 µL was injected for a GC analysis.

The homogenates of the spinal cord in PBS (0.5 mL) were prepared according to the following procedure. In brief, each sample was added to tridistilled water (1 mL) and chloroform/methanol (6 × 4 mL). The organic layers were collected, dried over anhydrous Na2SO4 and evaporated under vacuum to dryness. The lipid extract residues (1.0 ± 0.05 mg), consisting of phospholipids and cholesterol, as revealed using TLC (n-hexane-diethyl ether 9:1), were converted to FAMEs and analyzed following the procedures described for RBCs. FA values were calculated as relative percentages of quantitative analyses (% rel. quant.) and expressed as mean ± SD.

### 2.4. Macrophage (RAW 264.7) Cell Line and In Vitro Inflammation

The macrophage cell line RAW 264.7 (ATCC^®^ TIB-71™) was purchased from the American Type Culture Collection (ATCC^®^ TIB-71™). Cells were plated in glass coverslips onto a 24-well culture plate and grown in DMEM high-glucose medium (Thermo Fisher Scientific, Waltham, MA, USA) supplemented with 10% heat-inactivated fetal bovine serum (FBS—Thermo Fisher Scientific) and 1% penicillin/streptomycin (100 U mL ^−1^/100 µg mL ^−1^) (Thermo Fisher Scientific) at 37 °C in a humidified incubator of 5% CO_2_.

To test the effect of the inflammatory stimulus on the cell membrane dynamics of the macrophages, we exposed the RAW 264.7 cell line to 500 ng/mL of lipopolysaccharide (LPS) for 4 or 8 h, a condition effective at inducing an inflammatory phenotype, as described elsewhere [27].

### 2.5. Primary Cultures and In Vitro Challenges

Cortical neurons from the fetal brains of C57BL/6J mice at 13.5 days postcoitus (E13.5) were prepared according to a standard protocol [28]. In brief, the fetal heads were placed in a Petri dish containing PBS 1× with 1% penicillin/streptomycin (100 U × mL^−1^/100 µg × mL^−1^). Using a stereotaxic microscope, the brains were removed using forceps and placed upright on the plate. Using forceps, the meninges were carefully removed, the olfactory bulbs removed and the cerebral cortex collected in a 1.5 mL tube containing a nonenzymatic dissociation buffer (Sigma-Aldrich, St. Louis, MO, USA). After 15 min of incubation at 37 °C, the tissues were pipetted several times for mechanical dissociation. After 5 min of centrifugation at 400× *g*, the cellular pellets were resuspended in a neurobasal culture medium supplemented with 1× B27 (Thermo Fisher Scientific), 2 mM glutamine (Sigma-Aldrich) and 1% penicillin/streptomycin (100 U × mL^−1^/100 µg × mL^−1^) (Thermo Fisher Scientific) and then plated onto Cultrex 2D substrate (0.25 mg/mL, Trevingen)-coated 96-well plates. The cells were maintained in a humidified incubator at 37 °C with 5% CO2. To obtain a pure neuronal culture (99% neurons), the cells were treated after 24 h with 5 μM cytosine arabinofuranoside (Sigma-Aldrich). No mitotic inhibitor was used for mixed cultures. Half of the medium was changed every 2–3 days.

For the OPC-enriched culture, fetal NSCs were isolated following previously published protocols [29]. In the same way as the cortical neuronal culture, the fetal heads were placed in a Petri dish containing PBS 1× with 1% penicillin/streptomycin (P/S; Thermo Fisher Scientific) (100 U × mL^−1^/100 µg × mL^−1^). Under a dissection microscope, the brains were removed using a lancet and placed upright on the plate. Using forceps, the meninges were carefully detached, the olfactory bulbs removed and the forebrains collected in a 1.5 mL tube.

After the nonenzymatic dissociation, the pellet containing the cell fraction was resuspended in a serum-free NSC medium (DMEM/F12 GlutaMAX; 8mmol/L HEPES; 100 U × mL^−1^/100 µg × ml^−1^ penicillin/streptomycin; 1 × B27; 1 × N-2; 10 ng/mL bFGF; 10 ng/mL EGF; Thermo Scientific) and the cells plated at a density of 10 cells/µL in a T25 flask (Corning, New York, NY, USA) following cell count. The flasks were kept in a vertical position to avoid cell adhesion and the medium was changed every three days. Neurospheres were allowed to proliferate until they reached a diameter of approximately 100 µm. To obtain the OPC-enriched spheres, the cells were centrifugated at 400× *g* for 5 min and resuspended in 1 mL of medium. The pellet containing the spheres was mechanically dissociated through pipetting; then, the cells were counted and plated again at a density of 10 cells/µL in OPC medium (DMEM/F12 GlutaMAX 1×; 8 mmol/L HEPES; 100 U × mL^−1^/100 µg × mL^−1^ penicillin/streptomycin; 1× B27; 1× N-2; 20 ng/mL bFGF; 20 ng/mL PDGF; Thermo Fisher Scientific). Following cell count, the cells were plated at a density of 3000 cells/cm^2^ on a poly-D,L-ornithine (50 µg/mL)/laminin (5 µg/mL; Sigma-Aldrich) coating in OPC medium.

To induce oligodendrocyte differentiation and maturation, the OPC medium was substituted with the oligodendrocyte differentiation medium (DMEM/F12 GlutaMAX 1×; 8 mmol/L HEPES; 100 U × mL^−1^/100 µg × mL^−1^ penicillin/streptomycin; 1× B27; 1× N-2; 50 nM T3; 10 ng/mL ciliary neurotrophic factor (CNTF); 1× N-acetyl-L-cysteine—NAC; Thermo Fisher Scientific) following 3 DIVs.

To test the response of the neuron plasma membrane to different insults, cultures were exposed to LPS and an inflammatory cytokine treatment. As described for the RAW 264.7 cell line, the LPS treatment was performed with a concentration of 500 ng/mL for 8 h. To test the cytokine mix recapitulating the pathological features of EAE on OPCs in vitro [30], we used the same conditions to test the effect of the inflammation on neurons, the OPC-enriched culture and OL-enriched culture by exposing the cultures for 24 h to a mix consisting of TGF-𝛽1, TNF-𝛼, IL-1b, IL-6, IL-17 and IFN-γ (10 ng/mL), as the cytokine mix was described as the main trigger of the OPC differentiation block in the EAE model [31].

### 2.6. Lipid Probes and Microscopy

For the Filipin III (Sigma-Aldrich, ID number F4767) staining, the cells were washed twice in PBS 1× and fixed in a 4% paraformaldehyde solution for 15 min at room temperature. After rinsing in PBS 1×, the cells were incubated with 25 μg/mL of Filipin III for 1 h at room temperature [32]. The cultures were then washed 3 times in PBS 1× before imaging.

For the cholera toxin B subunit (CTX-B—FITC conjugate) (Sigma-Aldrich, ID number C1655) staining, the cells were washed twice with PBS 1× and incubated with 2.5 μg/mL of CTX-B for 30 min at 4 °C (protected from light). After incubation, the cells were rinsed 3 times with cold PBS 1× and fixed in a 4% paraformaldehyde solution for 10 min at room temperature. For the CT-B staining, the cultures were also washed 3 times in PBS 1× before imaging.

RAW 264.7 cells cultured on glass coverslips were mounted on microscopy slides in 0.1% glycerol/1,4-phenilendiamine (Sigma-Aldrich) and analyzed with fluorescence microscopy using a Nikon Eclipse E600 microscope equipped with a digital CCD camera Q Imaging Retiga-2000RV (Q Imaging, Surrey, BC, CA) and Nis-Elements AR 3.2 software. This software was also used for the fluorescence analysis, drawing a region of interest (ROI) on each cell to quantify the fluorescence emitted due to the Filipin III staining.

The primary cortical neurons, OPCs and OL-enriched culture cultured in 96-well plates were analyzed using a cell-based high-content screening (HCS) technology (Cell Insight NXT; Thermo Scientific). For the acquisition and analysis, a dedicated software was used (HCS Studio v 6.6.0, Thermo Fisher Scientific), selecting the “cell morphology” algorithm that detected the cell body based on CT-B fluorescence, measuring its intensity.

### 2.7. RBC Scan Electron Microscopy

SEM was performed to assess whether the pathology had an impact on the RBC morphology. RBCs were isolated from whole blood using Ficoll-Paque ™ PLUS (Cytiva, cat. number 1714402) at a ratio of 1:1, centrifuging at 3000 rpm for 20′ at room temperature (RT). After washing in 1× PBS (Carlo Erba Reagents, ID number MS00V11001) and diluting at a ratio of 1:10, the RBC pellet was fixed with 2.5% glutaraldehyde in phosphate tampons for 2 h at RT. The pellet was then incubated in a phosphate buffer. A SEM analysis was performed using a Zeiss EVO 40 electron microscope (Zeiss, Oberkochen, Germany).

The fluorescent staining of the RBC lipid membrane and blood smear procedure were conducted to determine whether the disease affected the membrane lipid content; three fluorescent stainings were performed: (1) 6-dodecanoyl-2-dimethylaminonaphthalene (hereinafter referred to as Laurdan) (Merck, Sigma-Aldrich, cat. number 40227) as an indicator of membrane fluidity; (2) cholera toxin B subunit (CT-B, Merck, Sigma-Aldrich, ID Number C1655), whose target was the ganglioside-monosialic acid (GM1) within the lipid rafts, with CT-B being an indirect reporter of esterified cholesterol, one of the most prominent components of lipid rafts; (3) the Filipin complex from Streptomyces filipinensis (hereinafter referred to simply as Filipin) (Sigma-Aldrich, ID number F9765), which measured the free (nonesterified) cholesterol.

The staining procedure followed the same steps for all the probes, with the exception of the incubation concentrations, times and temperatures.

To completely remove all plasma components which might negatively affect the staining, the RBCs were isolated from whole blood using Ficoll, as previously described, after which the pellet was fixed with 2.5% glutaraldehyde in a phosphate buffer for 30′ at RT. After removing the glutaraldehyde, and to reduce fixative autofluorescence, the RBC pellet was incubated with 1.5 mg/mL glycine (Glycine, PlusOne™, purchased by Fisher Scientific from Life Science products Cytiva, cat. number 17-1323-01) for 30′ at RT. In total, 15 µL of RBCs was then stained using Laurdan 25 µM (from a stock solution of 5 mM) in the dark at RT for 2 h, 1 ug/mL CT-B (from a stock solution of 1 mg/mL) in the dark at 4 °C for 30′ and 0.05mg/mL Filipin (from a stock solution of 25 mg/mL) in the dark for 2 h at RT.

After washing the pellets in PBS (3 ×; 2000 rpm, 5′, RT), the RBCs were diluted in PBS 1:3. A total of 3 µL of RBCs was then smeared on a 1mm thick glass slide (Menzel Gläser, Thermo Scientific, ID number ABAA000001##12E) using a coverslip oriented at a 45° angle. Very thin one-layer smears were prepared to avoid any cell clustering, which would have negatively affected the image acquisition.

### 2.8. Fluorescent Microscopy and Postprocess Imaging

Images were acquired using a Nikon Eclipse Ci-S upright microscope (Nikon, Tokyo, Japan) with a PlanApo 100×/1.40 NA objective with a working distance of 0.13 mm in immersion oil type F. The exposure time and gain were set to allow for a comparison among all samples, with the images acquired in both greyscale and false color. Laurdan-stained slides were imaged as described in Wenzel et al., 2018 [33], and the image postprocessing was performed using the ImageJ software (https://imagej.nih.gov/ij/ (accessed on 16 September 2020)). The postprocessing steps were different for CT-B, Filipin and Laurdan. To determine the differences between healthy and pathological mouse RBCs, the fluorescence intensity was calculated for CT-B and Filipin. The Laurdan results were expressed as generalized polarization (GP), calculated using the GP calculator macro [34].

The grayscale images were deconvoluted for the CT-B and Filipin analyses. This was performed by, firstly, calculating the point spread function (PSF) using the Diffraction PSF-3D ImageJ plugin (PSF—https://imagej.net/plugins/diffraction-psf-3d (accessed on 06 June 2005)), and then by calculating the deconvolution using the DeconvolutionLab ImageJ plugin, as described in Vonesch and Unser 2008. After deconvolution, the membrane pixels were identified (using the Find Edges option) and greyscale attribute filtering was applied to each image (Gray Scale Attribute Filtering ImageJ plugin in MorphoLibJ library, https://imagej.net/plugins/morpholibj (accessed on 14 May 2020)) to apply a gray value to each pixel of the RBC membrane.

Finally, the fluorescence intensity was measured by calculating the corrected total cell fluorescence (CTCF), subtracting the integrated density from the product of the selected area and obtaining the mean fluorescence of the background readings.

The pseudocolored images were also postprocessed using ImageJ to remove the background and identify membrane pixels.

For the Laurdan evaluation, after finding the membrane pixels (using the Find Edges option), the images were processed using the GP calculator macro (https://sils.fnwi.uva.nl/bcb/objectj/examples/CalculateGP/MD/gp.html (accessed on 10 June 2016)) to remove the background and create two pseudoimages (termed A and B), which were then combined into a single image with false colors. The result was shown using a color LUT red–white–blue; three conditions were possible: (a) if red was predominant, image B would dominate; (b) if blue was predominant, image A would dominate; and (c) white areas would indicate a balanced condition. For this study, it was assumed that image A referred to a 440 nm emission, whereas image B referred to a 540 nm emission. A color bar with normalized values was also included as a reference, ranging between −1 and +1. The color range between −1 and 0 indicated a reduced membrane fluidity, whereas the color range between 0 and +1 suggested an increased membrane fluidity.

### 2.9. Statistical Analysis

Data were reported as mean ± SEM. The Prism software PRISM (v 6.0 or 8.0.1; GraphPad Software, Boston, MA, USA) was used for the statistical analyses and graph generation. The statistical analysis was based on Student’s *t*-test for a comparison between two groups; an ANOVA and post hoc test for a comparison between more than two groups; a nonparametric unpaired t-test for the biochemical analysis. The results were considered significant when the probability of their occurrence as a result of chance alone was less than 5% (*p* < 0.05).

## 3. Results

### 3.1. The EAE Mouse Model Reflects Remittent Neuroinflammation and Demyelination

The clinical progression of EAE was monitored via body weight and a neurological disability score (Figure 1A,B). The body weight in the immunized mice diverged from the control group at 10 DPI, remaining constantly lower thereafter (Figure 1A, group effect *p* < 0.05). Neurological deficits appeared at 10–15 DPI, with a rapid increase in disability; most mice attained the highest clinical score at 15–20 DPI (the acute phase), with a peak in severity at approximately 20 DPI, followed by a partial and stable remission until 50–55 DPI (the remission phase).

The spinal cord was the CNS region most severely affected by neuroinflammation in active immunization EAE [35,36]; therefore, we focused the histopathological evaluation of this area (inflammation and demyelination) on the two investigated times (acute: 20DPI; remission: 50 DPI) using toluidine blue staining to visualize the inflammatory infiltrate, markers for demyelination (MBP), astroglial reaction (GFAP) and axonal integrity (NF200) (Figure 1C–N). As expected, the toluidine staining showed a perivascular (closed arrows), submeningeal (open arrows) and intraparenchymal (asterisk) infiltrate of inflammatory cells at 20 DPI, a time at which extensive myelin disaggregation (MBP-IR, Figure 1d,h,l, high-magnification inserts) and axonal loss (NF200, Figure 1f,j,n, high-magnification inserts) were observed. While the myelin integrity was partially recovered at 55 DPI, the axonal loss was stable over the observational time. The astroglial reaction, as visualized with GFAP-IR in the dorsal funiculus (df) (Figure 1e,i,m, high-magnification inserts), was prevalent at 50 DPI. We also investigated Iba1-IR as a marker for microglial activation and neuroinflammation, showing a strong activation at 20 DPI that persisted at 55 DPI (see Appendix A). Iba-1-positive cells took on the expected phagocytic morphology (large cell body and retracted elongation, Appendix A). Panels D–F show the negative controls obtained by omitting the primary antisera.

### 3.2. Fatty-Acid-Based Membrane Lipidome Analysis of RBCs and Spinal Cords

Blood was obtained from the mice at the three phases of the study: (1) the starting time, defined as the control stage (CTRL); (2) the acute phase of the disease (EAE); and (3) the remission phase of the disease (REM). Following the work-up for membrane isolation and transformation to FAMEs, the fatty-acid-based membrane lipidome profile was determined, as previously described [37,38].

The FAME values in the RBC membranes are shown at the three experimental times in Figure 2 and Appendix A. The omega-6 and omega-3 PUFA family components were significantly decreased in the acute phase of EAE; the long-chain omega-3 PUFA DHA in particular was significantly reduced, together with the omega-6 arachidonic acid. While PUFA could have been incorporated into membrane phospholipids by adding PUFA precursors to the diet, changes in PUFA could not be explained with diet in our case, given that all the mice were fed the same diet (diet composition is shown in Appendix A). Moreover, when omega-6 and omega-3 changes occur in membranes in favor of the former, it is well known that the proinflammatory predisposition of cell metabolism increases. Indeed, the FA indexes, such as omega-6/omega-3 and ARA/DHA ratios, also significantly diminished in the EAE phase (Figure 2, Appendix A), indicating that not only was the overall polyunsaturated content of the membranes reduced, but that their pro- and anti-inflammatory balance also changed, as shown with the RBC analysis. Notably, the EAE phase was characterized by a significant increase in the SFA palmitic acid, indicating the activation of lipogenesis in the EAE mice. In combination with the PUFA loss, the SFA increase also suggested a change in membrane properties, which warrants further investigation.

We were gratified with the results of the RBC membrane lipidome analysis in the subsequent remission (REM) phase. As shown in Figure 2, the remission phase was mainly accompanied by a return to the control levels of SFA and PUFA residues. In the latter case, the PUFA level restoration was largely due to increases in omega-6 linoleic and DGLA levels. Interestingly, the monounsaturated FA (MUFA) levels were found to increase in the REM phase compared to both the CTRL and EAE phases, whereas the SFA stearic acid decreased.

Finally, we wanted to examine, at least preliminarily, the effects of EAE induction on the FA profile of nervous tissue. We obtained samples of spinal cords by sacrificing a small number of mice (*n* = 3 for each time point) in the CTRL and EAE statuses, and Table 1 shows the SFA, MUFA, omega-6 and omega-3 PUFA levels in the two conditions. It was gratifying to see that some of the FA markers found in RBC membranes were equally modified in the spinal cord; we also noted a MUFA increase in the remission phase, thus, confirming the important contribution of the MUFA metabolism to nervous system health. Decreases in DHA, which influences the total omega-3 level and increases the omega-6/omega-3 ratio, were also significant, reflecting the changes in the RBC profile.

### 3.3. Erythrocyte Morphology and Membrane Lipid Probes during EAE

RBCs from the same experimental animals used in the FA analyses were examined in parallel for their morphology and membrane properties. RBCs imaged with SEM and derived from experimental animals are shown in Appendix A, where (a) shows the control RBCs, and (b) and (c) show EAE RBCs in the acute and remission phases of EAE, respectively. No morphological differences in the three conditions were appreciable.

We then examined the lipid composition and fluidity in the entire cell membrane using fluorescence probes. Figure 3A shows representative RBCs stained with CT-B and the image analysis resulting from the fluorescence intensity evaluation (B). RBCs (zoomed-in mode) representative of the control (CTRL), acute and remission phases of EAE are shown in (a), (b) and (c), respectively; (d), (e) and (f) show the postprocessing images, in which only pixels belonging to the plasma membrane are shown. The image postprocessing flow chart is given in Appendix A. The control RBCs were brighter than the RBCs in the EAE acute phase and a restoration to the control labeling was observed in the remission phase. The fluorescence intensity results (B) representing the mean of the CTCF measured in 30 RBCs under control, acute and remission conditions, respectively, were expressed as a CTCF value (arbitrary unit—A.U.) and confirmed image observations.

Figure 3C shows representative RBCs stained with Filipin and the image analysis staining intensity evaluation (D). Representative RBCs (zoomed-in mode) collected at the control, acute and remission phases of EAE are shown in (a), (b) and (c), respectively; (d), (e) and (f) show the postprocessing images, in which only pixels belonging to the plasma membrane are displayed. RBCs have a very high nonesterified cholesterol content in the acute phase, when compared with cells at the control and remission stages, as indicated by brighter cells at early EAE compared to the control animals and confirmed with quantification (D), expressed as a Filipin value (A.U.).

Figure 3E shows representative RBCs stained with Laurdan, and the image analysis procedure for the staining intensity evaluation (F). The image postprocessing flow chart is shown in Appendix A. Representative RBCs (zoomed-in mode) collected at the control (CTRL), acute and remission phases of EAE are shown in (a), (b) and (c), respectively; (d), (e) and (f) show the postprocessing images, in which green and red colors were chosen to differentiate between the two emissions, where the green color shows the more fluid area and the red color the more rigid membrane area. For the GP calculation, NAN (not a number) pixels were excluded and only the membrane pixels were considered. Figure 3F shows the ratios between the saturated and unsaturated FAs, thus, displaying the return to control conditions during the remission phase.

### 3.4. Plasma Membrane Lipids of Macrophages Dynamically Respond to Inflammatory Stimuli

To test whether direct exposure to inflammatory stimuli would alter lipid distribution in the plasma membrane, we exposed the RAW 264.7 cell line to an LPS-mediated inflammatory insult. RAW 264.7 are murine monocyte/macrophage-like cells described as an appropriate and popular model of macrophages, highly reactive to inflammatory stimuli [39]. We used the same conditions (LPS 500 nM) already described to trigger macrophage activation at two different exposure times (4 and 8 h) [27].

To analyze the cell membrane lipid changes, we used two probes, as in the RBC experiments: Filipin, which binds to plasma membrane cholesterol, and CT-B, which binds GM1, a ganglioside associated with lipid rafts, which acts as a lipid receptor (Figure 4A). In cultures not exposed to LPS, the staining did not change at culture times of between 4 and 8 h (Appendix A) and were, therefore, combined in a unique control group. At 4 h of LPS treatment, cells showed an increase in Filipin staining (one-way ANOVA, F(2.34) = 7.356; *p* = 0.0022; Dunnett post-test, *p* = 0.0010), while no differences compared to the control group were found at 8 h (Figure 4B–E).

CT-B staining was also affected by LPS exposure (one-way ANOVA, F(2.34) = 3.975; *p* = 0.0284), with a decrease in the fluorescence observed at 8 h (Dunnett post-test, *p* = 0.0470) (Figure 4F–I).

### 3.5. Inflammatory Challenges Acutely Affect Plasma Membrane Lipid Composition in Primary Cortical Cells

To explore whether the plasma membrane of neural cells reacts to noxious stimuli active during neuroinflammation, we analyzed lipid probes in neuronal and oligodendroglial cultures challenged with LPS or a cytokine mix [40] as a model of in vitro inflammation.

We initially characterized the pure (99% neurons) and mixed (50% neurons and 50% astrocytes) cultures for CT-B staining. A previous characterization showed these cultures to contain only these two cell types [41], easily distinguishable by the different nucleus size and cell morphology. The “cell morphology” module of the HCS system shows the cell area or perimeter delimited with the CT-B staining, while nuclear staining was used to avoid the selection of debris. The ramified structure of the neurons, which develops from small cell bodies and nuclei, is shown as a separated structure from the underlying layer of astrocyte flat and wide cells. The CT-B staining clearly marked the neurons while remaining weak in the astrocytes, allowing for the easy separation of the two cell types based on the fluorescence threshold. However, using the same threshold value, the pure neuronal cultures had a lower GM1 content, as shown by weaker CT-B staining (Appendix A), and may reflect the higher vulnerability described for the pure neuronal culture [28], which mimicked the in vivo conditions less accurately, prompting us to choose the mixed system to test the pathological challenges (Figure 5A).

Both LPS and cytokine exposure evoked an increase in Filipin staining compared to the control cultures (one-way ANOVA, F(2.11) = 4.986; *p* = 0.0287; Dunnett’s post-test, LPS, *p* = 0.0359; cytokines, *p* = 0.0281) (Figure 5B–E), while a modification in CT-B staining was shown with the cytokine treatment only, with a strong reduction (one-way ANOVA, F(2.12) = 13.36; *p* = 0.0009; Dunnett’s post-test, *p* = 0.0006) (Figure 5F–I).

OPC-enriched cultures were obtained by driving NSC differentiation using specific factors (Figure 6A–J). Primary neurospheres were obtained in the presence of EGF and bFGF, and following neurosphere splitting, EGF was replaced with PDGF in the culture medium, triggering the differentiation of the OPCs (OPC-enriched spheres). To obtain the OPC-enriched cultures, the spheroids were dissociated, seeded as single cells and cultured for three DIVs (Figure 6A). At this stage, the bFGF/PDGF medium was replaced with a medium containing T3, allowing for OL maturation (Figure 6J). An analysis was performed by creating a mask on the NG2 and MBP-positive cells to identify the OPCs and OLs, respectively, using the “cell morphology” module of the HCS system. Comparing NG2-positive cells to the control culture condition, only cytokine exposure induced an increase in fluorescence intensity for both Filipin (one-way ANOVA, F(2.9) = 16.94; *p* = 0.0009; Dunnett’s post-test, *p* = 0.0031) (Figure 6B-E) and CT-B staining (one-way ANOVA, F(2.9) = 12.78; *p* = 0.0023; Dunnett’s post-test, *p* = 0.0014) (Figure 6F–I). MBP-positive cells, corresponding to mature OLs, showed an increase in the fluorescence of Filipin staining for cytokines and LPS (one-way ANOVA, F(2.9) = 17.19; *p* = 0.0008; Dunnett’s post-test, LPS, *p* = 0.0005; cytokines, *p* = 0.0089) (Figure 6K–N), but no changes in CT-B staining (Figure 6O–R).

## 4. Discussion

Membrane lipids constitute the bulk of the dry mass of the CNS, and the relative lipid composition and packaging influences membrane fluidity and neural cell physiology. Lipids also make up 70–85% of myelin [42], which consists of cholesterol and glycolipids, at a percentage ratio of 40:40:20 (cholesterol, phospholipids and glycolipids, respectively), as opposed to most biological membranes, where the ratio is 25:65:10 [3]. Lipidomics, therefore, represent an emerging tool for research into demyelinating diseases such as MS, an attractive perspective as a biomarker and a target for dietary intervention.

There is still, however, a lack of clarity regarding the relationship between the membrane lipid composition of peripheral and CNS cells, such as the impact of circulating lipids on the CNS lipid composition. Indeed, while cholesterol is synthetized in situ by astrocytes and oligodendrocytes under physiological conditions [43], FAs can easily cross the blood–brain barrier (BBB) via passive diffusion or transcytosis [44,45].

To explore a number of aspects of the complex interplay between peripheral cells and the CNS in the lipid pathways, we analyzed RBCs and the FA composition of spinal cord tissue in the EAE mouse model, the most widely used animal model of inflammatory/demyelinating disease. We also investigated the impact of inflammation on lipidic microdomains that also reflect membrane fluidity in RBCs and cells, which play a key role in MS pathology, i.e., macrophages, neurons, OPCs and OLs.

The FA composition was investigated during the acute and remission phases of EAE; the former characterized by severe systemic and CNS inflammation and a breakdown of the BBB, the latter through myelin repair attempts [29,33]. FA pools in EAE animals were altered in a very specific way, with changes that are reflected in the RBC membrane lipidome. The SFA–MUFA metabolism, which is determined by the endogenous enzymatic formation of palmitic acid (16:0), its elongation to stearic acid (18:0) and its desaturation to oleic acid (9c-18:1), showed an accumulation of SFA and a MUFA decrease during the acute phase of EAE (cfr., Figure 2). This effect may be due to the inhibition of stearoyl-CoA desaturase (SCD-1), evoked as a key step in cell death also [46], thereby influencing the biophysical status of the membrane toward rigidity.

PUFA families of omega-6 and omega-3 were also significantly reduced in acute EAE, returning to control levels at the remission phase, especially the two omega-6 linoleic and dihomo gammalinolenic (DGLA, 20:3) acids. It is well known that cell membranes undergo continuous phospholipid remodeling in a process known as Lands’ cycle, and our results indicated that membrane remodeling in the EAE remission phase occurred with the restoration of the unsaturation of MUFA and PUFA. Although the degree of saturation influences membrane properties, the membrane is the crucial site for lipid signaling; the arachidonic acid levels and omega-6/omega-3 ratio did not appear to change in the RBCs over the course of EAE, suggesting an interesting behavior of membrane-derived inflammatory markers, warranting further investigation.

The observed remodeling process had an important metabolic significance, given that the diet consumed by the mice did not change during the experiment. The experimental animal diet was based on an omega-6/omega-3 precursor ratio (linoleic/alpha-linolenic acid ratio) of 8.8; however, the total omega-6/omega-3 ratio was 5.2 due to the presence of long-chain PUFA, EPA and DHA (Appendix A). It might, therefore, be expected that omega-6 would have been metabolically balanced by omega-3 under our experimental conditions, as described for the animal diet with the omega-6/omega-3 ratio lower than nine [47], which did not occur. Another important element described in [48] is the calculated dietary PUFA balance (omega-3/PUFA ratio) of 14.7 (Appendix A), which ensures that all types of tissues (in our case, RBCs and CNS tissues) receive a balanced lipid intake. The changes that we observed during the three phases were, therefore, a direct consequence of the health status induced by EAE. Notably, our previous results demonstrated that diets supplemented with EPA delayed the onset of clinically severe disease regulating FOXP3 expression, increased the expression of myelin proteins and improved the integrity of the myelin sheath, also polarizing peripheral lymphocytes [47].

It is remarkable that the RBC membrane monitoring throughout the course of the disease could be so informative of changes at the metabolic level. The loss of PUFA (the omega-3 DHA in particular) was accompanied by an increase in the lipogenesis metabolite palmitic acid, with dramatic consequences for the membrane properties. The monitoring of the fatty-acid-based lipidome, in fact, combined with the experiments on RBC membrane properties, such as CT-B, Filipin and Laurdan staining, demonstrated that the EAE phase coupled FA remodeling with membrane property changes. By using the same samples at the corresponding times for both the lipidome and fluorescent analyses, we were able to associate the FA changes with variations in membrane properties, as discussed in the next section, such as increased palmitic acid levels coupled with an increase in cholesterol levels, as detected with Filipin staining in the EAE phase (Figure 4D). In the remission phase, on the other hand, MUFA and PUFA (palmitoleic, oleic and vaccenic acids, and the omega-6 linoleic acid and DGLA) residues increased, providing a metabolic marker that linked disease remission with an increase in lipid unsaturation through the activation of desaturase enzymes. At the same time, the membrane properties monitored via Filipin staining during the remission phase detected a reduction in cholesterol levels (Figure 4D). It is worth noting that the palmitic acid levels were the same during the acute and remission phases of EAE (cfr., Figure 2), whereas the increase in unsaturation during the remission phase combined with the decrease in stearic acid to create a membrane environment which led to a fall in cholesterol. Palmitic acid increases in membranes occur in other metabolic conditions, such as glucolipotoxicity, and has been associated with altered membrane fluidity parameters, as shown by various researchers [49].

These results offer a starting point for further experiments to investigate the contribution of FAs to membrane plasticity to improve cell fate outcomes and those of the body as a whole. The membrane lipid relationship described in our study also represents, to the best of our knowledge, the first description of a peripheral cell (RBC) over the entire course of EAE, from onset to remission, suggesting that the lipidic composition of peripheral cells may represent an important tool and a potentially comprehensive biomarker for the study and monitoring of the role of metabolic and dietary factors in the evolution of degenerative diseases.

In this study, we also provided preliminary indications that the FA profile of RBCs may potentially reflect that of the CNS; in fact, some of the FA markers found in RBC membranes were equally altered in the spinal cord, such as the DHA decreasing (cfr., Table 1), while the ARA/EPA and ARA/DHA ratios significantly increased, possibly due to a greater inflammatory predisposition of the tissue or a decreased availability of FAs, known to be precursors of bioactive lipids with neuroprotective properties. On the other hand, the MUFA oleic acid increased, which was an interesting result given that oleic acid is deemed to be a neurotrophic factor for spinal cord injury [50]. The limited number of spinal cord samples included in the study did not permit any further insights into the connections between the RBC lipidome and the tissues of the CNS.

These results may also have a translational importance for MS. Low PUFA consumption and an inadequacy of CNS FA elongation and desaturation have been reported in MS patients for decades. As the major components of the OL membrane, systemic and resident CNS FA levels fluctuate significantly during demyelination, such as FA metabolism-related genes and proteins (reviewed in Yu et al., 2022). FA composition in RBCs also changes in MS patients, revealing an increase in shorter long-chain saturated FAs, while higher levels of C14:0 and C16:0 reflected better disease outcomes, as demonstrated by the inverse correlation with the expanded disability status scale and functional system score [51,52].

Overall, the indications for the RBC lipidome as a noninvasive diagnostic marker for CNS, being easy to use as a follow-up during disease evolution, were preliminarily gathered, suggesting that a FA-based RBC membrane lipidome could serve as a “personalized tool” in clinical applications [53,54].

Our study also aimed to identify new biomarkers that could act as cellular sensors to provide a simple, rapid, innovative and noninvasive tool for diagnosis, prognosis and possibly therapy monitoring. RBC plasma membranes are two-dimensional structures, consisting of a cytoskeleton and a lipid bilayer tethered together [55], with equal proportions by weight of cholesterol, phospholipids and integral membrane proteins. It is well known that cholesterol is distributed equally between the two leaflets, whereas the four major phospholipids are asymmetrically arranged [56]. Recent studies have described “lipid rafts” rich in cholesterol and sphingolipids in association with specific membrane proteins [57] and gangliosides.

The RBC membrane is highly flexible. Moreover, according to a mechanism that is not yet fully understood, it appears that RBCs may exchange lipids with bilayers from other cells [58,59]. We used Filipin staining, widely used as a histochemical marker for cholesterol in lipid membranes, and CT-B staining, the most well-established and popular probe of lipid rafts binding GM1 ganglioside, the key component of lipid domains, also directly involved in the response to noxious stimuli, and which indirectly explores the cholesterol inside lipid rafts [60,61]. Our fluorescent staining-based results showed an alteration in the lipid content of RBC membranes related to disease progression, although their morphology did not appear to be modified by the disease. The decrease in GM1 during the acute phase of EAE reflected the breakdown of the lipid rafts, leading to an increase in the rigidity of the RBC membrane, as demonstrated by the Filipin and Laurdan staining. Compared to the control and acute phases, the RBCs from the remission phase showed a slight restoration of the cholesterol content, both in lipid rafts (CT-B staining) and free lipids (Filipin staining). Overall, the membrane fluidity was also closer to that of the control phase. Decreased fluidity in the RBC membranes in the EAE acute phase could be interpreted according to the hypothesis of cholesterol metabolism reprogramming in macrophages, as suggested in cancer cells [62]. RBC cholesterol metabolism may be reprogrammed with the use of intracellular signals activated by neurons and astrocytes involved in EAE progression and acting as paracrine signals.

These results open a very relevant clinical translational perspective. In fact, the access to CNS molecular composition is critical in humans, and clinicians are searching for a peripheral “shuttle” to act as CNS feedback. RBC may, therefore, represent a good candidate as a peripheral sensor of CNS conditions.

Noteworthy is that the ultrastructural images did not show appreciable modifications, highlighting the relevance of assessing alterations in membrane lipid composition to allow for the earlier detection of disease development and progression. It is remarkable that such lipid metabolic changes affecting the fluidity and permeability properties of RBC membranes can be detectable with fluorescence probes, in agreement with previous observations [49].

To directly explore the effect of inflammation on the lipid composition and membrane fluidity of cells involved in EAE, we performed in vitro experiments using the same lipid probes as in the RBC experiments. We initially analyzed macrophages, a key cell in driving EAE pathophysiology, which crosses the BBB and acts as a phagocyte with resident macroglia when activated [34,63]. At 4 h after LPS exposure performed at 37 °C, the increased Filipin staining suggested a cholesterol increase in the macrophage membrane; upon a longer exposition to LPS, the Filipin staining reduced, presenting together with a reduced presence of GM1, as indicated by the decreased CT-B staining. While no functional conclusions could be drawn from these data, it should be borne in mind that an increase in the membrane fluidity has previously been associated with the phagocytic phase of macrophages [64]. The membrane fluidity of macrophages was reduced in the M1 proinflammatory morphology, and increased when the macrophages switched to the M2 type, with anti-inflammatory and prophagocytosis characteristics [65,66].

We performed the same experiments on mixed cultures of neurons/astrocytes exposed to LPS and cytokines, observing the same membrane lipid profile changes in neurons under inflammation stimuli, showing an increase in Filipin and a decrease in CT-B staining. It should be noted that our mixed in vitro systems recapitulated the in vivo conditions, in which astrocytes play a fundamental role in exchanging lipids with the other cells, as described in oligodendrocytes [67]. This cell–cell interaction in fact reflected the different cholesterol contents identified in neurons cultured with or without astrocytes.

However, a different lipid profile was observed in the cells responsible for myelination; Filipin staining increased in both OPCs and OLs exposed to inflammatory challenges, thus, suggesting that cholesterol in the plasma membrane was modified as in the other cell types investigated. On the contrary, and at the investigated time-points, CT-B increased in OPCs, but not OLs. Thus, we could speculate that the cholesterol content increases in OPCs and OLs during inflammatory conditions, possibly impacting membrane fluidity.

The results obtained in both neurons and oligodendroglial cells in terms of the cholesterol composition of their cell membrane reflected the fundamental role of this lipid in physiological processes and in response to inflammation and demyelinating insults. OL membranes, in fact, contain a high amount of cholesterol, which is required for membrane compaction [68]. Moreover, it has been described that cholesterol synthesis is essential for driving remyelination, both in neurons [69] and oligodendrocytes [70].

Gangliosides are abundant in mammalian neurons and play different roles in neuronal plasticity and the release of neurotrophins, as well as in cell differentiation, memory control, cell signaling, neuronal protection, neuronal recovery and apoptosis. Gangliosides are also receptors for different toxins, bacteria, viruses and autoantibodies [71]. GM1, which is mainly present in neurons, plays an important role in maintaining neuronal functions and plasticity [72], and different studies have shown a correlation between the alteration of GM1 in neurons and the progression of neuronal diseases [73,74]. Gangliosides are also crucial for the organization of the myelin structure, and it has been hypothesized that they are mostly abundant in oligodendrocyte membranes, compared to the other cell inhabitants, probably contributing to the definition of nodes and paranodes in myelinated fibers through myelin-associated glycoprotein–ganglioside binding, ensuring optimal axon–myelin cell–cell interactions [75].

GM1 plays different roles in the CNS to other gangliosides, interacting with all main cellular components; it has been shown to exert an anti-inflammatory effect on microglia [76] and stabilize lipid rafts, preventing myelin injury [77]. The different roles played by GM1 in different cell types may explain the diverse effects exerted by the same inflammatory stimulus on the various cells analyzed. In fact, it decreased or remained stable in neurons and mature OLs, the main cells vulnerable to inflammation in terms of viability, while increased in OPCs, in which inflammation induced migration and replication [31,40].

The analysis of FA composition was not performed in this in vitro experiment. In fact, the main aim was to perform a single-cell analysis using a culture system, at least mimicking the in vivo cell composition, as the mixed cultures and related milieu that we used.

## 5. Conclusions

In this study, we demonstrated that the lipid membrane composition can be altered by inflammation in RBCs, as indicated by the FA profile, cholesterol, GM1 and membrane fluidity probes investigated during the acute and remission phases of EAE, which indicated a more rigid profile. Notably, the FA profile in the spinal cord tissue resembled the FA alterations in RBCs, while in vitro exposure to inflammatory challenges altered the cholesterol and GM1 membrane content in macrophages, neurons and oligodendroglial cells in a similar manner that was mirrored by RBCs during acute EAE. Changes in FA composition, cholesterol and GM1 lipid raft probes also reflected an increase in stiffness during acute inflammation. Results from this preclinical study performed in murine models suggest that the lipid membrane profile of peripheral cells should be further investigated as a potential minimally invasive biomarker of CNS neuroinflammation. The use of human samples in formal clinical studies and human cells for in vitro experiments would strongly increase the translational potential of our results.

## Figures and Tables

**Figure 1 cells-12-00561-f001:**
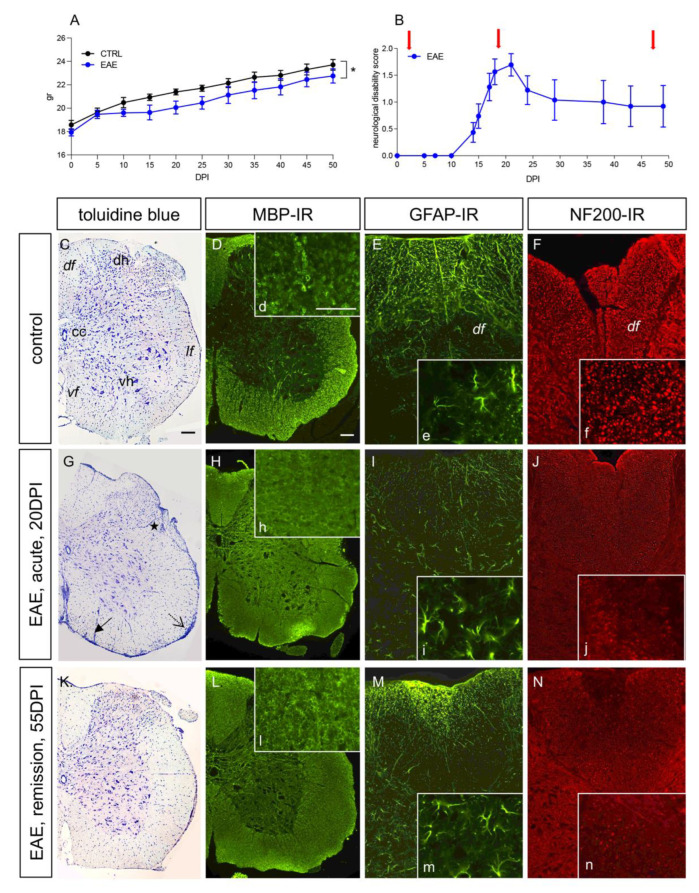
EAE clinical profile and histopathology. A,B. Body weight gain (**A**) and disability score (**B**) during the observational time. Data were expressed as mean + SEM. Statistical analysis: two-way ANOVA, * *p* < 0.05. (**C**–**N**). Histology (toluidine blue staining, **C**,**G**,**K**), MBP-IR (**D**,**H**,**L**), GFAP-IR (**E**,**I**,**M**) and NF-200-IR (**F**,**J**,**N**) in control mice (**C**–**F**), EAE acute phase (**G**–**J**) and EAE relapse phase (**K**–**N**). The inserts identified with lowercase letters show high magnification of the corresponding micrograph for MBP (**d**,**h**,**l**), GFAP (**e**,**i**,**m**), and NF200 (**f**,**j**,**n**) stainings; symbols in (**G**) indicate the perivascular (closed arrow), meningeal (dotted arrow) and parenchymal (asterisk) inflammatory cellular infiltrates. Scale bars: C (D–N) 100 mm; d (**e**,**f**,**h**,**i**,**j**,**l**,**m**,**n**) 50 mm. *Abbreviations: cc, central canal; df, dorsal funiculus; dh, dorsal horn; lf, ateral funiculus; vf, ventral funiculus; vh, ventral horn*.

**Figure 2 cells-12-00561-f002:**
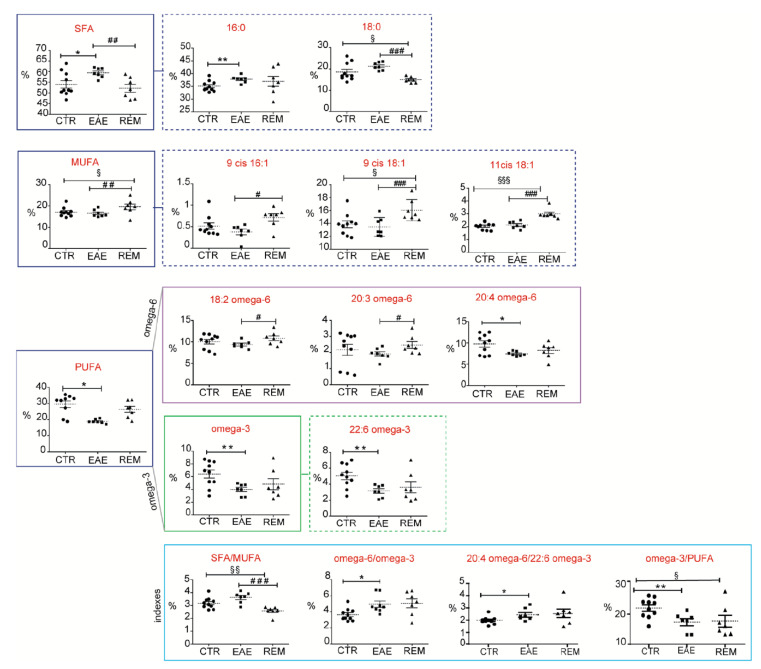
FAs in RCBs during EAE. Scattered dot plots of the statistically significant FAs and indexes of RBC membranes (see Table 1 for values); statistics (unpaired *t*-test) report on the comparison of: EAE vs. CTR; *: *p* value ≤ 0.045; ** *p* value ≤ 0.0097; REM vs. EAE: # *p* value: 0.048; ## 0.009; ### 0.0005; REM vs. CTR: § *p* value: 0.04; §§: 0.008; §§§ ≤ 0.0001.

**Figure 3 cells-12-00561-f003:**
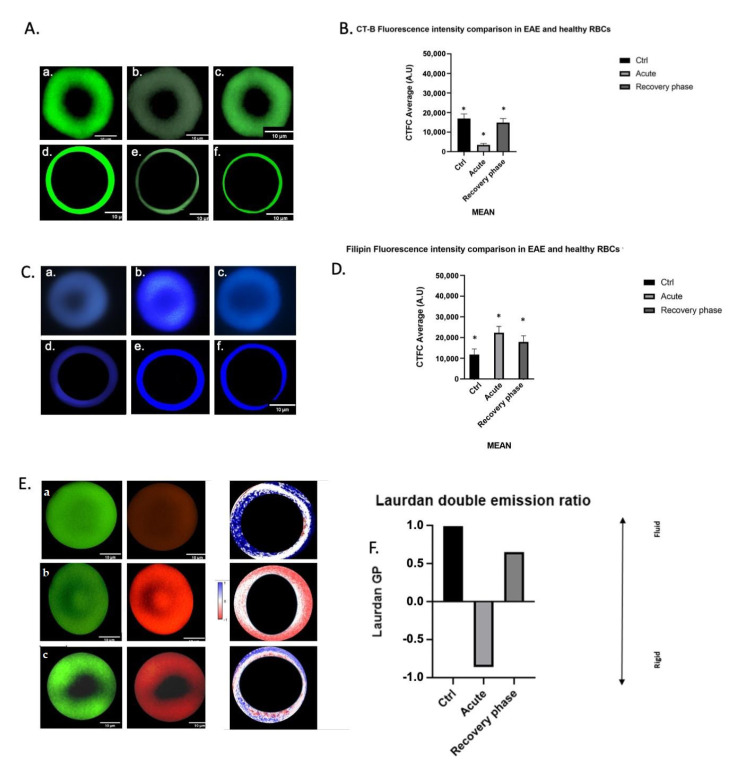
Lipid probes in RBC plasma membranes during EAE (**A**). CT-B fluorescence in healthy and EAE mouse RBCs (**a**–**c**) and postprocessed images obtained with ImageJ (**d**–**f**). Scale bars: 10 µm. (**B**). CTFC fluorescence intensity values in healthy and EAE mice during acute and remission phases (mean + SD, sample size *n* = 30; statistical analysis: Kruskal–Wallis nonparametric test, where * *p* < 0.05). (**C**) Filipin fluorescence in healthy and EAE mouse RBCs (**a**–**c**) and postprocessed images obtained with ImageJ (**d**–**f**). Scale bars: 10 mm. (**D**). Filipin fluorescence intensity values in healthy and EAE mice during the acute and remission phases (mean + SD, sample size *n* = 30; statistical analysis: Kruskal–Wallis nonparametric test, where * *p* < 0.05). (**E**). Laurdan fluorescence values in healthy (**a**) and EAE mouse RBCs (**b**), acute; (**c**), remission, where green and red colors were chosen to differentiate between the two emissions. The corresponding green/red channel ratios (GPs) for each condition are also presented. The colored bar on the upper right side of the images indicates the rate of membrane fluidity, ranging from -1 (in blue), the lowest level of fluidity, to +1 (in blue), the maximum level of membrane fluidity, with 0 (in light red/white) being the medium level of fluidity. (**F**). The graph shows the green/red channel ratios of Laurdan fluorescence intensity values in the control, acute and remission phases of EAE (mean + SD, sample size *n* = 30).

**Figure 4 cells-12-00561-f004:**
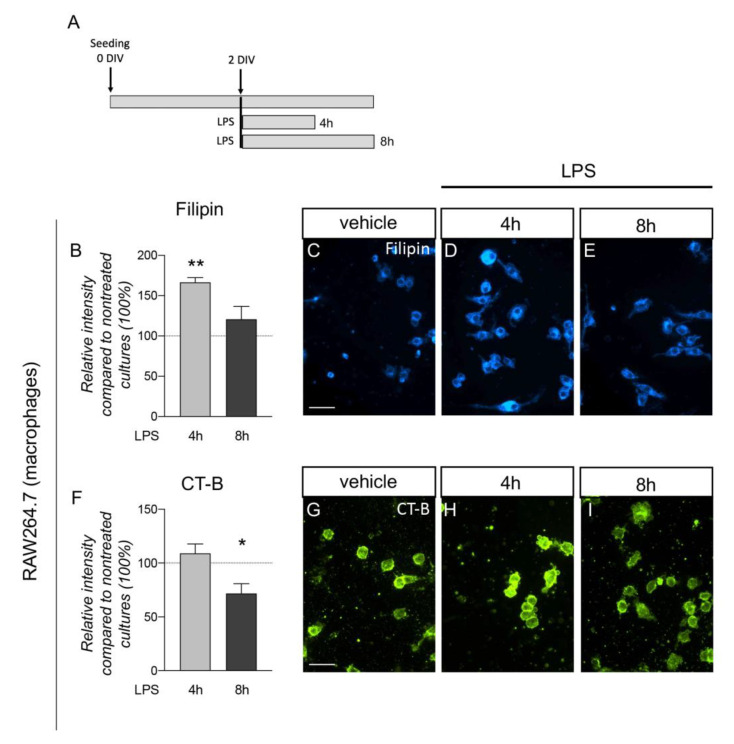
Lipid probes in macrophage plasma membranes exposed to LPS. (**A**). Schematic representation of the experimental protocol. RAW 264.7 cells were seeded on coverslips and cultured 2 DIVs before treatment. Cells were stopped after 4 and 8 h. (**B**). Graph shows the relative intensity of Filipin staining compared to nontreated cultures (100%, horizontal dotted line) in macrophages (RAW 264.7). Bars represent mean value + SEM. Statistical analysis: one-way ANOVA; asterisks represent differences compared to control cultures (** *p* < 0.01). (**C**–**E**). Representative images of macrophages (RAW 264.7) treated with vehicle (**C**) or exposed to LPS for 4 (**D**) and 8 (**E**) hours (h). Scale bar: 50 µm. (**F**) Graph shows the relative intensity of CT-B staining compared to nontreated cultures (100%, horizontal dotted line) in macrophages (RAW 264.7). Bars represent mean value + SEM. Statistical analysis: one-way ANOVA; asterisks represent differences compared to control cultures (* *p* < 0.05). (**G**–**I**). Representative images of macrophages (RAW 264.7) treated with vehicle (**G**) or exposed to LPS 4 h (**H**) and LPS 8 h (**I**). Scale bar: 50 µm.

**Figure 5 cells-12-00561-f005:**
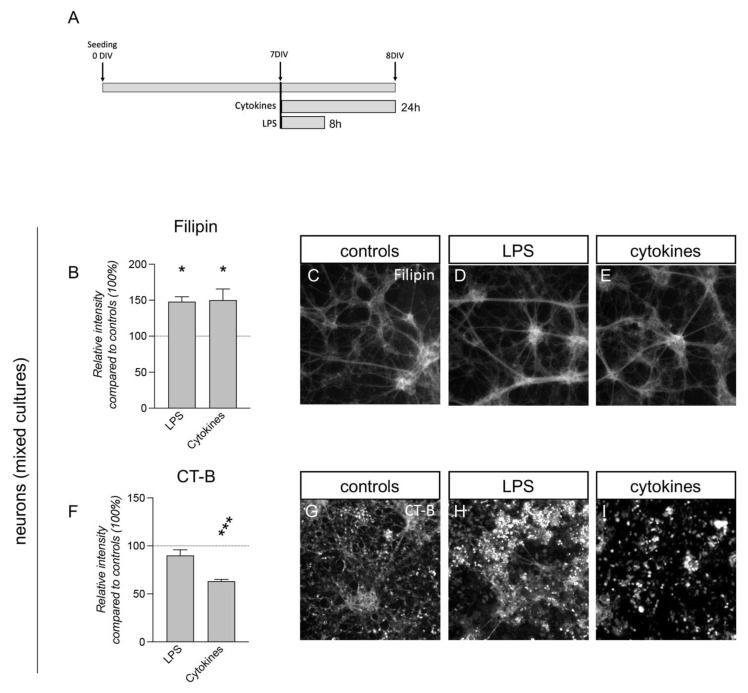
Lipid probes in neuron plasma membrane exposed to LPS and inflammatory cytokines. (**A**). Schematic representation of the experimental protocol. Cortical neurons were isolated from fetal brains at 13.5 days postcoitus (E13.5) and seeded on 96-well plates with Cultrex precoating. (**B**). Graph shows the relative intensity of Filipin within the fluorescence channel masked by the β-III-tubulin marker, compared to controls (100%, horizontal dotted line) in neuron/astrocyte mixed cultures. Bars represent mean value + SEM. Statistical analysis: one-way ANOVA; asterisks represent significant differences compared to control cultures (* *p* < 0.05). (**C**–**E**) Representative images acquired with cell-based HCS of neuron/astrocyte mixed cultures treated with vehicle (controls, **C**) and exposed to LPS (**D**) or cytokines (**E**). (**F**) Graph shows the relative intensity of CT-B staining compared to controls (100%, horizontal dotted line) in neuron/astrocyte cultures. Bars represent mean value + SEM. Statistical analysis: one-way ANOVA; asterisks represent significant differences compared to control cultures (*** *p* < 0.001). (**G**–**I**). Representative images acquired with cell-based HCS of neuron/astrocyte mixed cultures treated with vehicle (controls, **G**) and exposed to LPS (**H**) or cytokines (**I**).

**Figure 6 cells-12-00561-f006:**
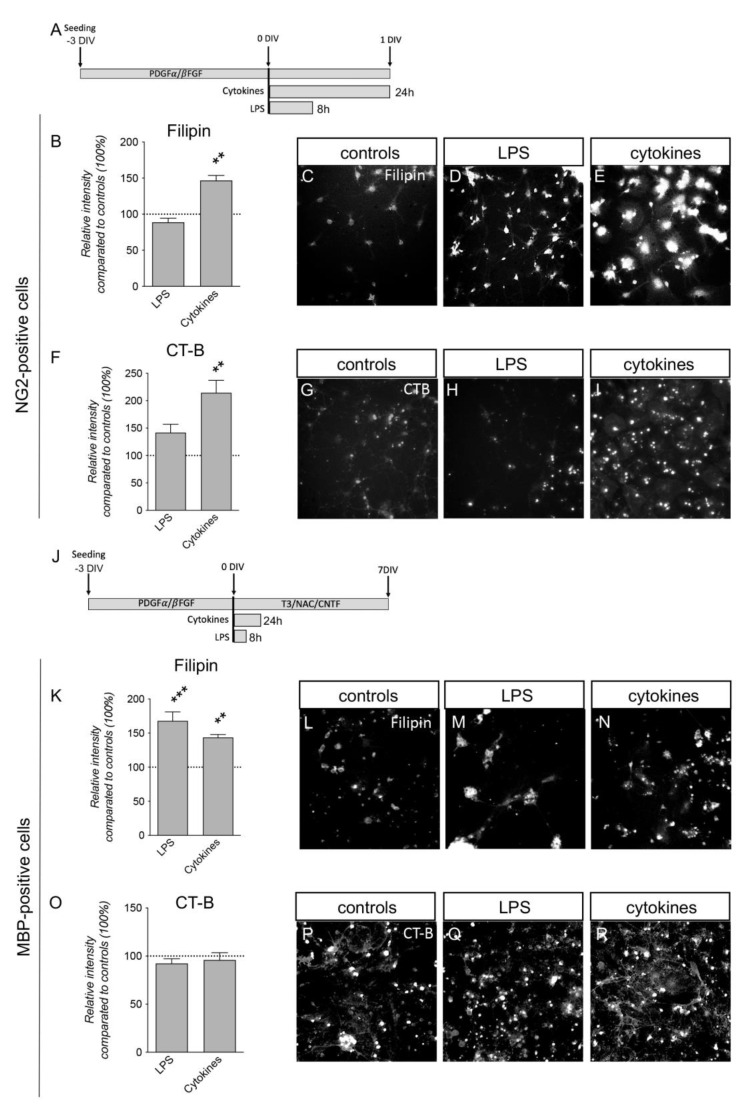
Lipid probes in OPC and OL plasma membranes exposed to LPS and inflammatory cytokines. (**A**). Experimental protocol. Cells derived from oligospheres were seeded in 96-well plates on poly-L-ornithine and laminin coating and treated after 3 days. (**B**). Graph shows the relative intensity of Filipin staining in NG2-positive cells (controls = 100%, horizontal line). (**C**–**E**). Representative images from cell-based HCS of NG2-positive cell controls (**C**) and exposed to LPS (**D**) and cytokines (**E**). (**F**). Graph shows the relative intensity of CT-B staining in NG2-positive cells (controls = 100%, horizontal line). (**G**–**I**) Representative images acquired with cell-based HCS of CT-B-stained cells treated with vehicle (**G**, controls) and exposed to LPS (**H**) or cytokines (**I**). (**J**) Experimental protocol. Cells derived from oligospheres were seeded in 96-well plates on poly-L-ornithine and laminin coating and treated after 3 days. On the same day (0 DIV), oligodendrocyte differentiation was induced with the differentiation mix consisting of triiodothyronine (T3), N-acetyl-cystein (NAC) and CNTF. Mature oligodendrocytes were analyzed after 7 DIVs. (**K**). Graph shows the relative intensity of Filipin staining in MBP-positive cells (controls = 100%, horizontal line) at 7 DIVs after differentiation induction. Bars represent mean value + SEM. (**L**–**N**). Representative images acquired with cell-based HCS of cultures treated with vehicle (**L**, controls) and exposed to LPS (**M**) or cytokines (**N**). (**O**) Graph shows the relative intensity of CT-B in MBP-positive cells (controls = 100%, horizontal line) at 7 DIVs after differentiation induction. (**P**–**R**). Representative images acquired with cell-based HCS of cultures treated with vehicle (**P**, controls) and exposed to LPS (**Q**) or cytokines (**R**). Bars represent mean value + SEM. Statistical analysis: one-way ANOVA; asterisks represent differences compared to control cultures (** *p* < 0.01; *** *p* < 0.001).

**Table 1 cells-12-00561-t001:** FA components of the total lipids extracted from the spinal cord of mice at three different phases of the disease.

	Spinal Cord CTRL (*n* = 3)	Spinal Cord EAE (*n* = 3)
FAME ^1^ (% rel quant)	mean ± sd	mean ± sd
14:0	0.27 ± 0.03	0.40 ± 0.18
16:0	16.35 ± 0.36	17.00 ± 0.94
16:1 trans	0.01 ± 0.00	0.00 ± 0.00
(6 + 7) 16:1 cis	0.16 ± 0.01	0.43 ± 0.18
9 cis 16:1	0.50 ± 0.03	0.52 ± 0.13
18:0	15.86 ± 0.31	15.63 ± 0.19
9 trans 18:1	0.04 ± 0.05	0.04 ± 0.01
8 cis 18:1	0.02 ± 0.01	0.06 ± 0.03
9 cis 18:1	28.44 ± 0.92	32.35 ± 1.99 *
11 cis 18:1	5.66 ± 0.06	5.04 ± 0.24 *
5 cis, 8 cis 18:2	Traces	Traces
mtrans 18:2	0.01 ± 0.00	0.03 ± 0.03
18:2 omega-6	0.98 ± 0.17	0.71 ± 0.06
18:3 omega-6	0.11 ± 0.01	0.11 ± 0.02
18:3 omega-3	0.11 ± 0.03	0.07 ± 0.03
20:0	1.73 ± 0.04	1.67 ± 0.30
11 cis 20:1	7.54 ± 0.47	6.15 ± 1.12
13 cis 20:1	2.06 ± 0.04	1.71 ± 0.41
20:2 omega-6	1.25 ± 0.06	1.17 ± 0.37
20:3 omega-6	0.58 ± 0.04	0.58 ± 0.08
20:4 omega-6	6.47 ± 0.18	6.22 ± 0.16
mtrans 20:4	0.03 ± 0.01	0.04 ± 0.00
20:5 omega-3	0.04 ± 0.01	0.03 ± 0.00
13 cis 22:1	0.74 ± 0.03	0.69 ± 0.10
22:5 omega-3	0.31 ± 0.04	0.27 ± 0.00
22:6 omega-3	9.47 ± 0.42	8.08 ± 0.63 *
24:0	0.42 ± 0.08	1.07 ± 0.22 **
15 cis 24:1	0.26 ± 0.00	0.25 ± 0.03
SFA	34.63 ± 0.73	35.77 ± 0.88
MUFA	44.39 ± 1.14	46.25 ± 0.53
PUFA	19.32 ± 0.55	17.25 ± 1.01
omega-6	9.38 ± 0.14	8.80 ± 0.43
omega-3	9.93 ± 0.41	8.45 ± 0.65 *
omega-6/omega-3	0.95 ± 0.03	1.04 ± 0.05 *
SFA/MUFA	0.78 ± 0.03	0.77 ± 0.01
ARA/EPA	166.59 ± 4.64	251 ± 32.99 *
ARA/DHA	0.68 ± 0.02	0.77 ± 0.4 *
omega-3/PUFA*100	51 ± 1	49 ± 1
tot trans	0.08 ± 0.04	0.11 ± 0.04

^1^ Fatty acids were obtained as FAMEs (fatty acid methyl esters) following work-up of the total lipids extracted from the spinal cord homogenates, as described in the experimental part of this study. Values were reported as relative quantitative percentages of each FAME, recognized and calibrated with standard references in the GC analysis. Values were expressed as mean ± SD of the three replicates. Statistical analyses were performed using Graph Pad 6. Unpaired *t*-test*: EAE vs. CTRL; EAE vs. CTRL: *p* value: * ≤ 0.045; ** ≤ 0.0085.

## Data Availability

Raw data are available from the corresponding authors upon reasonable request.

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
