# Peer review of "Erythrocyte Plasma Membrane Lipid Composition Mirrors That of Neurons and Glial Cells in Murine Experimental In Vitro and In Vivo Inflammation"

_cells, 2023, doi:10.3390/cells12040561_

Round 1

Reviewer 1 Report

The authors present the results of their research in great detail, which allows you to trace the experiment. The topic is very interesting, it carries a diagnostic element - preliminary results suggest that certain changes in the nervous tissue can be predicted on the basis of lipidomic studies of red blood cells. A valuable element in the analysis of fatty acids is the consideration of the diet of the tested animals. Minor typographical errors do not affect the scientific value of the work.

Author Response

We thank the reviewer for this very positive comment. The typos have been corrected.

Reviewer 2 Report

The aim of the study was to analyze the lipid composition changes due to inflammatory conditions. They analyzed RBCs and spinal cord tissue via gas chromatography from mice affected by experimental allergic encephalomyelitis (EAE) in acute and remission phases.  They examined the cholesterol and the membrane content of Filipin and GM1 membrane assembly (CT-B), in EAE mice RBCs, and in cultured neurons, oligodendroglial cells, and macrophages exposed to inflammatory challenges.

They found that during EAE acute phase the RBC membrane showed an increase in SFAs and the omega-6/omega-3 ratios and a reduction in PUFA, followed by a restoration to control levels in the remission phase, in parallel with an increase in monounsaturated fatty acid residues. The decrease of PUFAs was showed also in the spinal cord. CT-B staining decreased and Filipin staining increased in RBCs during acute EAE,  which was also observed in cultured macrophages, neurons, and oligodendrocyte precursor cells exposed to inflammatory challenges. The authors suggest that the regulation of lipid content is an indication of increased cell membrane rigidity during the inflammatory phase of EAE and supports the investigation of peripheral cell membrane lipids as a possible biomarker for CNS lipid membrane concentration and assembly.

General comments:

This is a well-designed, well-written study looking at the lipid composition changes during inflammatory conditions.

Limitations of this study

1.   To enhance the translatability of this body of work, the authors should have also included human blood from patients in varying stages of allergic encephalomyelitis to at least examine the lipid composition to examine if the changes they observed in mice mirror that in human cells. In addition, in vitro, human cell lines would have added more value.

2.   It would have been interesting to see the total lipid composition changes measured via gas chromatography not only in RBC but in the neurons, astrocytes and macrophages, and oligodendroglial cells after exposure to inflammatory challenges.

3.   The title of this paper should indicate that these experiments were performed in mice. suggested title:

 Murine erythrocyte lipid plasma membrane composition mirrors that of neurons and glial cells in experimental in vitro and in vivo inflammation.

Author Response

General comments:

This is a well-designed, well-written study looking at the lipid composition changes during inflammatory conditions.

We thank the reviewer for this very positive comment.

Limitations of this study

  1. To enhance the translatability of this body of work, the authors should have also included human blood from patients in varying stages of allergic encephalomyelitis to at least examine the lipid composition to examine if the changes they observed in mice mirror that in human cells. In addition, in vitro, human cell lines would have added more value.

We agree with the reviewer’s suggestion. In fact, clinical studies are planned as follow-up of this preclinical study. The use of human cells will be also important. However, available primary human cells are much less characterized when exposed to inflammatory challenges, as in our study, making difficult the discussion of results. We have included a note in the conclusion section (line 884).

  1. It would have been interesting to see the total lipid composition changes measured via gas chromatography not only in RBC but in the neurons, astrocytes and macrophages, and oligodendroglial cells after exposure to inflammatory challenges.

Again, we agree in principle with the reviewer’s comment. However, this type of experiment would require the sorting procedure of the cell culture, altering the membrane stability of the cells also in terms of membrane lipidic composition. Alternatively, it is possible to use “pure” cultures, a cellular system further away from the in vivo condition. This point has been already discussed in presenting Figure S6C of supplementary materials (Line 611). However, we have added a comment on this point in the revised version of the main text (Line 869).

  1. The title of this paper should indicate that these experiments were performed in mice. suggested title:

 Murine erythrocyte lipid plasma membrane composition mirrors that of neurons and glial cells in experimental in vitro and in vivo inflammation.

As suggested by the reviewer, we included the term “murine” in the title: “Erythrocyte lipid plasma membrane composition mirrors that of neurons and glial cells in murine experimental in vitro and in vivo inflammation”.

Reviewer 3 Report

In the article submitted by Stanzani A. and Sansone A. (et al.) and entitled "Erythrocyte lipid plasma membrane composition mirrors that on neurons and glial cells in experimental in vitro and in vivo inflammation", the authors focus on the search of a non-invasive biomarker reflecting CNS status in inflammatory neurological diseases. Briefly, they found that lipid composition of RBCs membrane may reflects the status of spinal cord in mice, concluding that lipidome analysis might be an useful and non-invasive stategy to detect inflammation in CNS.

In my opinion, the study is accurate and described in a good structured writing. However, in some part you should mention how your work can be relevant for human patients and what is its power in clinical pratice.

Moreover, since RBC reflects the condition of CNS in mice models, have you ever thought performing the same analysis on RBCs isolated from human blood? Of course analysis on spinal cord from patients is hard, but I think that, when possible, the use of human samples adds more value to your research.

Thanks

Author Response

In my opinion, the study is accurate and described in a good structured writing. However, in some part you should mention how your work can be relevant for human patients and what is its power in clinical pratice.

We thank the reviewer for this very positive comment. Indeed, in the discussion we highlighted he importance of our findings in the murine model both for the use of RBC as reporter of CNS status (although we underlined the few samples of spinal cord examined) and for the translational importance (lines 806-824) for patient analysis, which will certainly be the target of future investigations.

Moreover, since RBC reflects the condition of CNS in mice models, have you ever thought performing the same analysis on RBCs isolated from human blood? Of course analysis on spinal cord from patients is hard, but I think that, when possible, the use of human samples adds more value to your research.

We agree with the reviewer’s suggestion. As also reported in the answer to reviewer 2, clinical studies are planned as follow-up of this preclinical study (line 884). At the moment, we have highlighted in the discussion previous results obtained by the authors (line 828, ref 53,54).